# Basic Science with Preclinical Models to Investigate and Develop Liquid Biopsy: What Are the Available Data and Is It a Fruitful Approach?

**DOI:** 10.3390/ijms23105343

**Published:** 2022-05-10

**Authors:** Benedetta Cena, Emmanuel Melloul, Nicolas Demartines, Olivier Dormond, Ismail Labgaa

**Affiliations:** Department of Visceral Surgery, Lausanne University Hospital (CHUV), University of Lausanne (UNIL), CH-1011 Lausanne, Switzerland; bennycena7422@gmail.com (B.C.); emmanuel.melloul@chuv.ch (E.M.); olivier.dormond@chuv.ch (O.D.)

**Keywords:** ctDNA, CTC, ctRNA, exosomes, circulating, biomarkers, precision oncology

## Abstract

The molecular analysis of circulating analytes (circulating tumor-DNA (ctDNA), -cells (CTCs) and -RNA (ctRNA)/exosomes) deriving from solid tumors and detected in the bloodstream—referred as liquid biopsy—has emerged as one of the most promising concepts in cancer management. Compelling data have evidenced its pivotal contribution and unique polyvalence through multiple applications. These data essentially derived from translational research. Therewith, data on liquid biopsy in basic research with preclinical models are scarce, a concerning lack that has been widely acknowledged in the field. This report aimed to comprehensively review the available data on the topic, for each analyte. Only 17, 17 and 2 studies in basic research investigated ctDNA, CTCs and ctRNA/exosomes, respectively. Albeit rare, these studies displayed noteworthy relevance, demonstrating the capacity to investigate questions related to the biology underlying analytes release that could not be explored via translational research with human samples. Translational, clinical and technological sectors of liquid biopsy may benefit from basic research and should take note of some important findings generated by these studies. Overall, results underscored the need to intensify the efforts to conduct future studies on liquid biopsy in basic research with new preclinical models.

## 1. Introduction

Deriving from prenatal testing [1,2], liquid biopsy—defined as the molecular analysis of tumor byproducts released into biological fluids such as blood—was ranked within the top-major breakthrough technologies [3]. Owing to its unique polyvalence, it rapidly raised a strong interest in translational and clinical research, to a point of being almost considered as the ‘holy grail’ in cancer management. Numerous applications were demonstrated including early diagnosis [4,5,6], detection of minimal residual disease [7], decision making for systemic treatments [8,9,10], prognostication [11,12] or even to understand complex biological traits of cancers [13,14,15,16]. The classical paradigm ‘from bench to bedside’ was not followed in this field, somehow skipping the step of basic science. Therewith, data on liquid biopsy from basic research are strikingly scant. In 2019, EACR-ESMO hold its first liquid biopsy-dedicated meeting where experts expressed their concerns regarding the lack of data in basic science: “A final observation, supported by virtually all the speakers, is that many questions relating to the biology of circulating tumor material remain unanswered. The need for robust, reproducible basic science must not be overlooked in the drive towards the development of clinical assays. Indeed, such research will be essential to the many attempts to harness liquid biopsy-based technologies” [17]. Certain underlying mechanisms involved in the release of circulating analytes have been deciphered. As an example, it is now known that apoptosis is an important source of circulating tumor DNA (ctDNA) [18]. However, the process of analytes release, such as DNA fragments, remains largely unknown, especially regarding their determining factors. Although it is implicitly assumed that DNA is released by all cancer cells in a similar manner, this idea is certainly reductive and inaccurate. As a consequence, the non-detection of ctDNA is usually interpreted as a false negative result related to a limitation of the technology rather than to a trait of the cancer biology, which is misleading. A recent study evaluating the analytical performance of 5 ctDNA assays showed that the likelihood of detecting ctDNA was modulated by multiple factors. In particular, mutations below 0.5% of variant allele fraction (VAF) were unreliably detected [19], a serious concern. In parallel and as an example, an experimental study using hepatocellular carcinoma (HCC) xenografts revealed the feasibility of detecting circulating mutations in plasma of mice and more importantly, identified a clone-dependent release of DNA fragments in this model harboring two different cell clones [20]. These studies exemplified that, despite its meteoric rise in research, liquid biopsy must still face many challenges. A better understanding of the fundamental mechanisms behind tumor byproducts release is a *sine qua none* condition to overcome these obstacles. Answering these questions will allow improving the technology for detection of circulating analytes and ultimately improving patients’ outcomes. Animal models in basic science offer valuable and unique opportunities to explore and mimic specific clinical scenarios. They may facilitate the deconvolution of complex processes, for which translational research with human samples is not tailored. This study aims to comprehensively review the available data on liquid biopsy in basic research for each circulating analytes (e.g., DNA, RNAs, cells and exosomes) and to discuss the relevance of this approach.

## 2. Circulating Tumor DNA (ctDNA)

Cell-free DNA (cfDNA) includes a variety of different DNA molecules found in body fluids. Schematically, cfDNA is released via two routes: an active release and a passive one. The latter seemed to predominate; it essentially relies on apoptosis and to a lesser extent on necrosis. Hence, apoptosis has been considered as the main source of cfDNA in both healthy and diseased individuals, during the last two decades. This vision is certainly an over-simplification of the complex biology underlying DNA release. In fact, at least eight different mechanisms of cfDNA release have been identified, to date [21]. 

In patients with cancer, cfDNA encompasses DNA fragments released by healthy tissue as well as by the solid tumor, the so-called ctDNA. CtDNA has been evidenced as a family of biomarkers that suscitated an extraordinary enthusiasm in research and in clinical oncology, due to its various applications and its capacity to accurately reflect genomic aberrations detected in the tumor [22]. Researchers have rapidly focused on how ctDNA may contribute to cancer management, testing its value for diagnosis, surveillance, prognostication and other applications. However, the fundamental biology of ctDNA has been overlooked. Consequently, there is a critical need to understand what factors determine the release of ctDNA into the bloodstream. In this light, preclinical models in basic research appear to be uniquely tailored to investigate several important mechanisms regulating ctDNA release. Data on this specific topic will be comprehensively reviewed hereunder. 

### 2.1. ctDNA in Basic Research at a Glance

First, studies on ctDNA in basic research are rare, with only 17 articles identified in the literature. Characteristics of these studies are detailed in Table 1. Briefly, a majority of studies were conducted in colorectal (*n* = 7) and lung (*n* = 5) cancers. Most studies used immune-compromised mice to generate xenograft models, either with intravenous, subcutaneous or orthotopic injection of cancer cell lines. Techniques of detection included RT-PCR and digital droplet PCR (ddPCR). Studies were divided based on their findings, in four subchapters: (I) assessment of ctDNA to monitor tumor burden (II) the origin of ctDNA, (III) the impact of the cell clone of origin and (IV) the determinants of ctDNA release.

### 2.2. ctDNA Recapitulates Tumor Burden 

Importantly, one must remember how fast the field of liquid biopsy has evolved during the last 15 years. Questions that may sound obvious today were not so evident back then. Whether ctDNA may reflect tumor burden is typically such a question. There have been compelling data evidencing this concept now, including from translational research. Notwithstanding, two studies using animal models had an important contribution in this field, in the early 2000s. These studies analyzed ctDNA to monitor tumor burden and assess response to therapies by generating xenograft models. In 2006, Kamat et al. used female athymic nude mice intraperitoneally xenografted with human ovarian carcinoma cells [23]. 

Animals were treated with docetaxel alone or combined with anti-angiogenic agents while ctDNA was quantified by real-time PCR targeting human specific *β*-actin. The level of tumor-specific DNA in the mouse plasma correlated with tumor burden and varied following chemotherapy treatment, showing a transient increase at 24 h, followed by a drop below baseline value. This suggested that ctDNA could be a useful biomarker to study response to treatments. Thereafter, Rago et al. published a pioneer study focusing on the relation between ctDNA level and tumor progression. The authors developed a high-sensitivity assay leveraging human DNA as a proxy of ctDNA in mice, and allowing to quantify ctDNA in small volumes of mouse plasma and offering in-life monitoring of systemic tumor burden [24]. Athymic nude mice were subcutaneously or intravenously xenografted with human colon cancer, colorectal adenocarcinoma or osteosarcoma and ctDNA was measured by real-time PCR targeting the human long interspersed nuclear element-1 (hLINE-1). A positive correlation between the levels of ctDNA in the bloodstream and tumor burden was demonstrated. Regarding the response to treatments, they developed a model of osteosarcoma cell line with modified thymidine kinase highly sensitive to ganciclovir to assess response to therapy. A potent therapeutic response was observed in the treated group, which showed an initial increase in hLINE-1, followed by a decrease below pre-treatment levels. This finding was not observed in the vehicle group. Impact of chemotherapy and surgery on ctDNA level was also investigated. A spike of ctDNA concentrations immediately after tumor resection was observed, followed by a rapid decline in all mice. Altogether, these results showed that ctDNA can be used to study neoplastic development in mice bearing human xenograft tumors, evidencing its value to monitor tumor progression and to predict response to therapies. 

As some cancers are driven by gene translocations (e.g., Ewing sarcoma is driven by the reciprocal translocation between *EWS* gene and *ETS* transcription factor (Fli-1)), a research group aimed to detect circulating DNA fragments carrying this translocation breakpoint to assess tumor burden. Authors were able to detect translocated DNA, using ddPCR, and showed that EWS-FLI1 breakpoint amount in plasma correlated with tumor progression and rapidly dropped after surgical resection [25]. In 2017, Zhao et al. established a patient-derived xenograft (PDX) model of papillary renal cell carcinoma (pRCC) harboring an activating mutation of *MET*. The study included a group treated with cabozantinib—a non-specific *MET* inhibitor—as well as a control group. CtDNA levels increased with tumor progression in the control group and reflected response to therapy in the treated animals [26]. 

### 2.3. ctDNA Is Mainly Released through Apoptosis

Numerous studies have now identified apoptosis as a major source of ctDNA [18], while again, things were not so clear a decade ago. Several studies using animal models contributed to this knowledge. DNA fragments’ length depends on the mechanisms of release. Apoptosis is characterized by an increased fragmentation, compared to necrosis or phagocytosis. Mouliere et al. conducted two studies on this topic. A first study focused on ctDNA fragmentation and has explored the importance of amplicon length in ctDNA quantification [27]. They demonstrated that ctDNA amplification was impacted by target length, showing optimal values for DNA fragments around 60–100 bp. These finding were validated in a second study showing that ctDNA was more fragmented than cfDNA [28]. A recent study provided valuable insights on apoptosis in ctDNA release. Docetaxel was utilized to trigger apoptosis and blood samples were collected 24 h and 48 h after treatment, in a model of small cell lung cancer (SCLC) [29]. Conclusions included two important points: (I) cfDNA concentration was amplified under apoptosis stimulation and (II) ctDNA detection showed its highest sensitivity 24 h after treatment. These findings may be translated to patients in order to optimize ctDNA detection, in the future. Finally, another study has focused on the origin of cell-free plasma DNA [31]. Mice were subcutaneously xenografted with different human tumor cells. Plasma samples of tumor bearing mice and control mice (tumor-free) were analyzed by quantitative real-time PCR targeting mouse- and human-specific *β*-actin sequences. A higher concentration of mouse-specific DNA was observed in xenotransplanted mice compared to the control group, suggesting that the increased plasma DNA concentrations found in cancer patients might originate from both tumor and non-tumoral cells. However, very interestingly, human-specific DNA (or ctDNA) was present at various levels in tumor bearing mice, depending on the implanted tumor cells type, suggesting that ctDNA release might be cell type dependent. 

### 2.4. ctDNA Release Is Clone-Dependent 

A recent study from our group analyzed whether liquid biopsy with ctDNA could capture intratumoral heterogeneity (ITH) [39] in mice xenografted with two different cell lines of hepatocellular carcinoma (HCC) (i.e., HuH7 and HepG2). Each cell line (clone) harbored specific mutations (apolipoprotein B [*APOB*] for HuH7 and fibrinogen alpha chain [*FGA*] for HepG2). Mice were divided in two groups: a group treated with sorafenib and a group receiving placebo. Using qPCR targeting hLINE-1 allowed confirming a correlation between ctDNA detection and tumor burden. In addition, clone-specific mutations were targeted by ddPCR. *APOB* mutation was not detected in any of the 21 mice, whereas *FGA* mutations was detected in six of the 20 mice, suggesting that HuH7 tumors shed lower amount of ctDNA into the bloodstream, compared to HepG2. Moreover, the impact of treatment on mutation detection was also analyzed, showing that *FGA* mutation was detected in six of the 10 mice treated with placebo, whereas it was not detected in any of the mice treated with sorafenib. This suggested that treatment impacts mutation detection but that the difference in ctDNA release between the different clones is independent of the pharmacologic factors since it was detected in the placebo group [20]. Altogether, qPCR and ddPCR results identified a clone-dependent release of ctDNA. This was also suggested by data from another study, where three different human colorectal carcinoma cell lines (HT29, LoVo and LS174T) were grown in culture and injected in nude mice [33]. DdPCR was used to assess the levels of DNAs released into the culture supernatants and mouse plasma. Interestingly, the authors found that non-housekeeping cell-line specific “gateways” operated on specific circulating analytes and may have opposite influences on a given readout in vitro and in vivo. In fact, DNAs were more intensively released from LoVo than LS174T cells in vitro, whereas the opposite occurred when these tumor cells were grown in vivo (LS174T ctDNA was more abundant in mice blood). Overall, these results suggested that ctDNA release is modulated by multiple factors including non-housekeeping control gateways, selectively regulating the release of DNA fragments into the bloodstream. Finally, another study has observed that the amounts of ctDNA released during tumor growth was cell-line specific [34]. This group of researchers utilized BALB/c nude mice xenografted subcutaneously with two different non-small cell lung cancer cell lines (H1975 and H460) and cfDNA was quantified using human LINE-1 and mouse *ACTB* genes by qPCR. A correlation between ctDNA and tumor weight was noted, but not with the mouse-derived cfDNA, which was instead constant during tumor progression. DdPCR targeting *KRAS* Q61H in H460 xenografted mice and *EGFR* T790M mutation in H1975 xenografted mice, identified a variation of ctDNA detection at similar tumor size, suggesting that the amount of ctDNA released during tumor growth may be specific to each cell line. 

### 2.5. Exploring the Determinants of ctDNA Release

Identifying the factors determining ctDNA release is as paramount as challenging. A study investigated whether *Stat3*, an oncogene involved in the development of metastatic breast cancer cells, affected the release of ctDNA in mouse blood [35]. Three models of BALB/c nude mice were orthotopically xenografted with mouse breast cancer 4T1 cells expressing different levels of *Stat3* (4T1-GFP control group, 4T1-Knockout-*Stat3* group and 4T1-Overexpress-*Stat3* group). *Stat3* expression correlated with tumor volume as well as with cfDNA detection. This is an important finding underpinning the impact of a given gene on the release of ctDNA. Rostami et al. explored the role of senescence in cfDNA release [36]. Results confirmed the contribution of necrosis and apoptosis in the release of cfDNA but nuanced the impact of the latter, which appeared relatively minor in some tumors. More importantly, the study revealed senescence has an important determinant capable to mitigate the release of cfDNA. A recent study generated animal models mimicking patients treated with radiotherapy and contributed to decipher some factors affecting cfDNA release. In a model of prostate cancer, the likelihood of detecting ctDNA varied according to the location of cancer. In their models, the authors detected a spike of ctDNA release upon radiotherapy in subcutaneously xenografted mice whereas this spike was not observed in animals bearing intratibial tumors [37]. 

## 3. Circulating Tumor Cells (CTCs)

CTCs existence and impact in the disease course of cancer patients have been widely acknowledged. Responsible for metastases development, CTCs endorse a major role in cancers and therefore a major role within the concept of liquid biopsy. Over the years, numerous studies have investigated CTCs with approaches being more and more sophisticated. The first step (and challenge) was to detect CTCs. Thereafter, CTCs were enumerated, and their number was correlated to prognosis. Finally, studies aimed to characterize CTCs and their different subtypes to better understand their impact on prognosis according to their specific features. Those initially included surface markers. Subsequently, top-notch technologies have allowed to analyze CTCs in more depth, for example with single-cell sequencing [40]. Data have even suggested a spatial heterogeneity of CTCs distribution according to the vessels where blood samples were collected, identifying an epithelial-to-mesenchymal transition (EMT), which would add an additional layer of complexity and challenge to CTCs-based liquid biopsies [41].

### 3.1. CTCs in Basic Research at a Glance

As for ctDNA, studies on CTCs in basic research are scant, with only 17 articles identified; those are summarized in Table 2. A majority of studies investigated breast (*n* = 9) and lung (*n* = 4) cancers. The data on CTCs in basic research will be comprehensively reviewed hereunder in three distinct chapters divided according to the preclinical models: (I) xenografts, (II) PDX and (III) CTCs-derived xenografts (CDX). 

### 3.2. CTCs in Xenografts

Xenograft models resulting from the injection (intravenous, orthotopic or subcutaneous) of cancer cell lines has been described to study CTCs. Of note, it is a particularly challenging option requesting the ability to detect CTCs in very small volumes of blood. Unsurprisingly, this approach has been rarely used by researchers, with only three identified studies. In 2008, Eliane et al. have tested different methods of blood collection and developed a performant assay (modified CellTracks system) to recover CTCs from breast cancer xenografts and PDX. CTCs were also analyzed in hepatocellular carcinoma (HCC) in a study of our group. A specific HCC cell line (i.e., GFP-miR-517a-Huh7) was utilized due to its aggressive behavior and metastatic potential. Cells were injected orthotopically while whole blood and organs were collected upon euthanasia to detect CTCs and metastases, respectively. Among 11 mice, 4 developed lung metastases. Of note, CTCs were only detected in these four animals whereas no CTC was captured in any of the seven non-metastatic mice. Moreover, this study demonstrated that the number of CTCs positively correlated with tumor volume, but not with the number or the diameter of HCC nodules [20]. The study by Wang et al.—previously mentioned for ctDNA—also tested the impact of *Stat3* expression on CTCs. As for ctDNA, *Stat3* was identified as a determinant of CTCs release, potentially regulated by Snail induction and associated with EMT changes [35]. 

### 3.3. CTCs in PDX

Patient-derived xenografts (PDX) result from the engraftment of tissue pieces of cancer (collected from resected specimens or tissue biopsies) in immune-compromised mice. These models display precious advantages and are thus frequently used in basic/translational research [57]. As for mice xenografted with cancer cell lines, the use of PDX for CTCs analyzes necessitate the ability to capture these CTCs in small volumes of blood. Although demanding, some reports showed the feasibility of this technique. The study by Torphy et al. exemplified the interest of this approach; the authors generated PDX models deriving from patients with pancreatic ductal adenocarcinoma (PDAC), one of the deadliest cancers. Mice were randomized in two groups: a group treated with an oral phosphatidylinositol-3-kinase inhibitor (BKM120) and a group receiving placebo. Whole blood samples were collected before and after treatment and submitted to a microfluidic chip targeting EpCAM to capture EpCAM+ CTCs, with the assumption that CTCs was a promising candidate to predict response to treatment in PDAC. Results supported this hypothesis, showing a decreasing CTCs number after BKM120 compared to placebo. In addition, isolated CTCs maintained their genomic aberrant driver *KRAS* mutation, identical to the original tumors. This supports the reliability and stability of the model [43]. 

Another study on colorectal cancer had a similar design and showed consistent results [44]. Giuliano et al. have investigated CTCs in BC-PDXs, their potential to induce metastasis and the influence of different treatments on CTC numbers and tumor burden. They demonstrated that BC-PDX models retain genomic, transcriptomic and proteomic profile of the original patient’s tumor and can provide a continuous and renewable source of human CTCs, suggesting that CTCs can be used to study the metastatic process and be evaluated to monitor molecular changes during tumor progression and drug resistance. They used 18 different PDX-bearing mouse models—16 obtained from patients without metastasis and 2 from patients with metastatic disease—and CTCs were detected from mouse blood by quantitative immunohistochemistry (IHC). Among the different PDX lines, 15 were positive for CTCs and a significant correlation between the presence of CTC clusters (but not with individual CTCs) and lung metastasis was observed. However, there was a variability in CTC number in different mice within the same PDX line, which might be due to intra-tumoral heterogeneity [46]. 

### 3.4. CTCs in CDX

CDX are somehow a subtype of PDX. Instead of engrafting a cancer tissue sample collected from patients, researchers can isolate CTCs from patients and further inject these CTCs in immune-compromised mice, giving raise to CDX in case of successful engraftment. This strategy has been used in a majority of studies (*n* = 10) exploring CTCs in preclinical models.

Bacelli et al. generated CDX to explore metastatic breast cancer. To do so, they utilized CellSearch—the only FDA-approved system—to recover CTCs that were further injected in NSG mice. The injection of <1’000 CTCs did not lead to metastatic growth within 15 months after implantation, whereas six mice receiving ≥1’109 CTCs developed multiple bone, lung and liver metastases within 6–12 months after implantation. Analyses identified a subpopulation of CTCs expressing EPCAM, CD44, CD47 and *MET*, thus including cancer cell stem markers. This subgroup of cells was referred as metastasis-initiating cells (MICs) and could represent an attractive target for the diagnosis and treatment of metastatic breast cancer [47]. In 2014, Yu et al. conducted a remarkable study which provided the proof-of-concept on the feasibility to generate breast cancer cell lines deriving from CTCs. This technical prowess permitted to thoroughly study each of the six cell lines and to assess their sensitivity to several drugs. These cell lines were injected in immune-compromised mice, revealing tumorigenic effect in 3/5 injected CTCs-derived cell lines [48]. In a study on triple-negative BC (TNBC), Vishnoi et al. showed that isolated CTCs induced liver metastasis in 66% of the injected animals, confirming the role of CTCs in metastases development. No metastasis was observed in other common sites for TNBC, such as brain, lung or spleen. Liver metastatic tissues were freshly processed and injected into another group of mice and CTCs analyses demonstrated an increased number of CTCs and tumor burden in the later generation of CDX. Finally, molecular analysis of CTCs allowed identifying a 597-gene signature recapitulating the risk of liver metastasis in TNBC [49]. 

CDX also appeared valuable in lung cancers. Most small cell lung cancers (SCLC) are inoperable, impairing the access to tissue samples and reinforcing the interest of liquid biopsy for these diseases. Hodgkinson et al. used CellSearch system to isolate CTCs from chemosensitive and chemoresistant patients, which were then injected into one or both flanks of immunocompromised mice. Palpable tumors were detected in four of six mice within 4 months after implantation and CDXs recapitulated the response to treatment and the genomics of the corresponding patient, highlighting the value of CDX to test and predict response to systemic therapies [51]. Drug resistance was also assessed with CDX in another study on SCLC, showing consistent results [52]. Another study chose CDX to test a novel drug combination in SCLC. A new regimen with a PARP inhibitor olaparib alone or in combination with the WEE1 kinase inhibitor AZD1775 was compared to the standard of cisplatin/etoposide in 10 phenotypically distinct CDX [53]. Of note, CDX were longitudinally generated, before treatment and upon disease progression. Treatment response to the new combination was heterogenous for both intensity and duration but tended to diminish when tested in animals generated upon progression. Genomic and proteomic analyses identified molecular predictor of response, such as *MYC* family members.

## 4. ctRNA and Exosomes

Data on ctRNA or exosomes are virtually inexistant with only two available reports. The study by Gasparello et al. discussed in the ctDNA section, also integrated miRNA. Three miRNAs were selected (i.e., miR-221, miR-222 and miR-141) and analyzed in vitro and in vivo, with colorectal cancer cell lines. As for DNA, results in supernatant and in plasma were opposite: HT-29 cells were associated with an increased release of miR-141 in vitro compared to LoVo and LS174T cells whereas it was the opposite in vivo [33]. Another study elegantly unveiled an important mechanism used by melanoma to escape immune system, using xenograft models [58]. Tumor cells upregulated PD-L1 expressed on the surface of released exosomes. 

## 5. Discussion

Although the lack of data on liquid biopsy in basic research was known upfront, this review allowed to thoroughly assess the available data and to provide a precise picture on the topic, confirming and even emphasizing how scarce these data are. The question is then to understand why. Certainly multifactorial, two reasons may, however, predominate: the attractivity of translational research and the fear of basic research. For the former, translational approach analyzing human samples is obviously appealing. The risk of generating data in animal models that may fail to be validated when translated to patients is by-passed. The field of liquid biopsy certainly estimated to save valuable time, efforts and resources including financial ones. For the latter, animal models necessitate working with very small volumes of blood and thus even smaller volumes of plasma. Many projects and ideas may have been aborted when considering this point and designing experiments. Initially, it may indeed sound utopic to recover CTCs or to detect circulating mutations in mice with a blood volume equivalent to one droplet.

The main contribution of this review was to highlight the relevance of conducting basic research on liquid biopsy. Not only feasible, the detection of circulating analytes in preclinical models gave raise to important discoveries and findings that could not be identified in translational research. These results are summarized in Figure 1. Again, the facts that liquid biopsy is able to reflect tumor burden and that apoptosis is a main source of ctDNA release are now widely acknowledged but studies in basic research discussed above had an important impact on this knowledge. Conversely, a pivotal observation was noted in studies from basic research, showing a clone-dependent release of ctDNA. If confirmed and validated, this may drastically impact the interpretation of all ctDNA-based liquid biopsies. The following question is to understand and identify which factors are determining the release of ctDNA. Basic research also provided data and leads on this point, identifying cell states such as senescence or gene (e.g., *Stat3*) as determinants of ctDNA release. Regarding CTCs, data have especially revealed the value of the different models deriving from CTCs in basic research, namely xenografts, PDX and CDX. Another elegant example was to generate stable cancer cell lines deriving from CTCs and offering a precious tool for in vitro studies.

Basic research also displays limitations. Therefore, it should not be considered as the perfect Ersatz of translational research but rather as a valuable alternative and complementary approach to better understand the biology and improve the development of liquid biopsy. Besides the limitations of blood volume already discussed, one major drawback is the immune interplay. Animal models essentially included immune-compromised mice. Hence, the role of immune cells and microenvironment in the release of circulating analytes is ignored by these models.

Perspectives-wise, basic research offers unique opportunities to further explore the clone-dependent release of circulating analytes, as well as to investigate other candidates factors that may impact these releases. In addition, there is a critical need to conduct studies on ctRNA and exosomes for which data were dramatically scarce. Of note, this review included data deriving from circulating analytes detected in the blood, but the concept of liquid biopsy is also applicable to other biological fluids such as urine, saliva, cerebrospinal fluid or bile. This is certainly an under-evaluated domain of liquid biopsy where research must also be further developed.

In conclusion, studies on liquid biopsy in basic research are rare but the available data demonstrated noteworthy pertinence and allowed significant discoveries and findings that could not have been identified with a translational approach. The efforts to conduct future studies and to generate new preclinical models exploring liquid biopsy should be urgently intensified. Of importance, basic and translational approaches are not in opposition. At the contrary, they may be complementary and synergize progress in the field. Hence, studies integrating both pre-clinical and clinical materials should be valued.

## Figures and Tables

**Figure 1 ijms-23-05343-f001:**
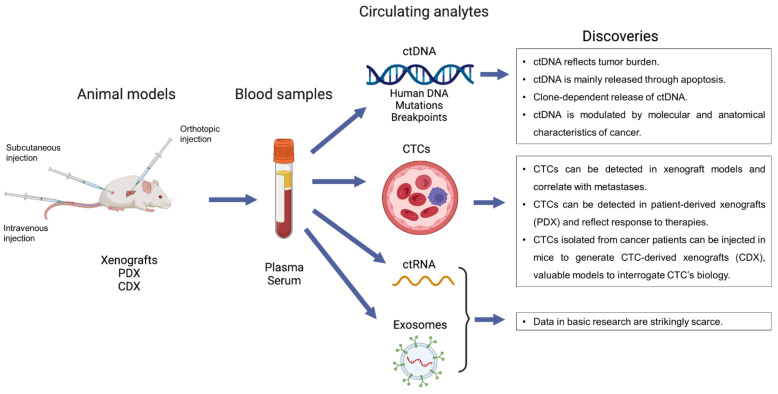
Schematic summary of the available studies on liquid biopsy in basic research with their main findings.

**Table 1 ijms-23-05343-t001:** Summary of studies investigating ctDNA in basic research.

Cancer Type(s) (Cell Line(s))	Animal Model(s)	Technique(s) of Detection	Readout(s)	Threshold (Volume of Blood)	Main Finding(s)	Reference
Human ovarian carcinoma (HeyA8)	Female athymic nude mice (orthotopic xenograft model)	qRT-PCR	Human specific *β*-actin	200 µL of plasma	cfDNA levels correlated with tumor burden. After chemotherapy, cfDNA showed a transient increase followed by a drop. The initial increase in cfDNA after treatment correlated with apoptosis.	[23]
Human colon cancer	Female athymic nude mice (subcutaneous xenograft model)	qRT-PCR	*hLINE-1*	100 µL of plasma	High-sensitive assay to quantify circulating human DNA (proxy of ctDNA) in small volumes of mouse plasma. CtDNA reliably reflected tumor progression, showing transient spike after cytotoxic or surgical treatments and a decrease after successful interventions.	[24]
Human colorectal adenocarcinoma (CL-188)	Female athymic nude mice (intravenous xenograft model)					
Human osteosarcoma (143B)	Female athymic nude mice (subcutaneous xenograft model)					
Human Ewing Sarcoma (TC71, EWS1, EWS4)	NSG mice implanted in the pretibial space with either TC71 xenografts or EWS1/4 PDX	ddPCR	*EWS-FLI1* tumor specific breakpoint	50 µL of whole blood	A sensitive assay was developed to detect breakpoint DNA fragments in xenograft and PDX models of Ewing sarcoma. CtDNA was detected and its concentration correlated with tumor burden, showing a significant decrease after surgical resection.	[25]
Human papillary renal cell carcinoma (pRCC) with MET mutation	Patient-derived xenografts (PDX) (orthotopic and subcutaneous engraftment)	qRT-PCR	*MET*	50 µL of plasma	A PDX model was established with pRCC tissue carrying a *MET* mutation. Mutant ctDNA was detected in plasma. Placebo group showed higher level of ctDNA compared to the group treated with Cabozantinib. Mice treated with cabozantinib showed decreasing amount of ctDNA upon treatment.	[26]
Human colon cancer (SW620, HT29)	Female athymic nude mice (subcutaneous xenograft model)	qRT-PCR	*KRAS* and *ACTB*	200 µL of plasma	CtDNA fragmentation increased with tumor size and tumor ctDNA concentration. CtDNA fragments showed a typical length of 60–100 bp.	[27]
Human colorectal cancer (SW620)	Female athymic nude mice (subcutaneous xenograft model)	qRT-PCR	*KRAS*	08–1.0 mL of whole blood	The proportion of mutant cfDNA varied but was overall quite high, suggesting that ctDNA account for an important part of cfDNA. In addition, ctDNA was more fragmented than non-tumor cfDNA.	[28]
Human small-cell lung carcinoma (H1975)	C57BL/6- Rag2^−/−^ IL2rg^−/−^ mice model (subcutaneous xenograft model)	qRT-PCR	*EGFR* mutation	160–600 µL of plasma	Docetaxel was used to induce apoptosis in a xenograft mice model harboring lung cancer. This treatment promoted apoptosis and facilitated the detection of ctDNA. Of note, the sensitivity of detection was maximal 24 h after treatment.	[29]
Human colorectal carcinoma (HCT116-s, SW620, HT29)	Female athymic nude mice (subcutaneous xenograft model)	qRT-PCR	*KRAS*, *BRAF*, *PSAT1*, *ACTB*	200 µL of plasma	In this xenograft model, non-tumoral circulating DNA remained low and constant whereas circulating tumor DNA (ctDNA) correlated with tumor progression. Circulating DNA features varied during cancer development.	[30]
Human lymphoblastoid cell (RPMI 1788)	Female BALB/c nude mice (subcutaneous xenograft model)	qRT-PCR	Mouse/Human *β*-actin	200 µL of plasma	Plasma samples from mice bearing human tumors contained human-specific DNA but also showed higher concentrations of mouse-specific DNA than control mice. This finding suggested that cancer development may also be associated with an increased release of healthy DNA fragments.	[31]
Human colon adenocarcinoma (KM12C, DLD-1)						
Human lung squamous cell carcinoma (SQ5)						
Human epidermoid carcinoma (A431)						
Human lung small cell carcinoma (SR-OV-3)						
Human breast cancer (MDA-MB-231, MDA-MB-468, KPL-4)	Athymic nude mice (subcutaneous and orthotopic xenograft models)	qRT-PCR	Human GAPDH, hLINE-1 and *AluJ*	200 µL of plasma	CtDNA was monitored during tumor progression in preclinical models treated with MEK inhibitor. CtDNA profile reflected tumor burden and response to therapy. Tumor size and levels of ctDNA were decreased in mice treated with MEK inhibitor, compared to controls. Importantly, the different cell lines showed varying ctDNA levels at similar tumor size.	[32]
Human colon cancer (Colo205)						
Human hepatocellular carcinoma (Huh7, HepG2)	Female athymic nude mice (subcutaneous xenograft model)	qRT-PCR and ddPCR	*APOB*, *FGA* and hLINE-1	0.5–1.0 mL of whole blood	Treatment with Sorafenib impacted mutation detection. Altogether, qRT-PCR and ddPCR results suggested a clone-dependent release of ctDNA into the bloodstream.	[20]
Human colorectal carcinoma (HT29, LoVo, LS174T)	Nu/CD1 nude mice (subcutaneous xenograft model)	ddPCR	*KRAS* (G12D and G13D)	2 mL of plasma	Cancer cell lines showed different release of ctDNA in vitro compared to in vivo. Altogether, results suggested that ctDNA release is modulated by multiple factors including non-housekeeping control gateways, selectively regulating the release of DNA fragments into the bloodstream.	[33]
Human non-small cell lung cancer (NCI-H1975, NCI-H460)	BALB/c nude mice (subcutaneous xenograft model)	ddPCR	hLINE-1 and *ACTB*, *KRAS* and *EGFR*	Not available	The concentration of ctDNA, but not non-tumor DNA, was positively correlated with tumor weight in both models. The fragmentation and detection rates of *EGFR* and *KRAS* mutations in plasma cfDNA increased along with ctDNA concentration and tumor weight. Moreover, H1975 and H460 xenografts showed varying ctDNA levels at similar tumor sizes, suggesting that the amounts of ctDNA released during tumor growth may be specific to each cell line.	[34]
Mouse breast cancer (4T1)	BALB/c nude mice (orthotopic xenograft modelusing 4T1 cells expressing *Stat3* transcription factor at different levels)	qRT-PCR	GAPDH and B1	200 µL of plasma	Three breast cancer xenograft models were generated, with different expression of *Stat3*. Both ctDNA and CTCs were detected and their detection rates correlated with *Stat3* expression.	[35]
Human head and neck squamous cell carcinoma (HMS-001, Cal33, Vu147T)	NOD-Scid-Gamma and NOD-Rag-Gamma male mice (subcutaneous xenograft model)	qRT-PCR	hLINE-1	1 mL of whole blood	Treatment type and the time interval from the treatment exposure are key factors impacting cfDNA release and detection. In addition, senescence was identified as a novel determinant of cfDNA release.	[36]
Human non-small cell lung cancer (HCC-827, PC-9)						
Human prostate cancer (LNCaP)	Athymic nude mice (subcutaneous and bone xenograft models)	qRT-PCR	hLINE-1 and *Alu*	35 µL of whole blood	Standard methods (slide caliper and bioluminescence) were compared to liquid biopsy to assess tumor burden in metastatic prostate cancer xenograft models treated with radiotherapy. Of note, although most animals showed a transient increase in ctDNA after ionizing radiation, this spike was not observed in animals bearing intratibial tumors.	[37]
Human non-small cell lung cancer (H1299, H460, H1975, HCC827)	BALB/c nude mice (subcutaneous xenograft model)	qRT-PCR and ddPCR	hLINE-1, *NRAS* and *EGFR*	Not available	A lung cancer xenograft model treated with radiotherapy. Both cfDNA and ctDNA were monitored and circulating mutant DNA was detected. While cfDNA was unchanged at 6 Gy, ctDNA was increased after radiotherapy.	[38]

Abbreviations: cfDNA: cell-free DNA; ctDNA: circulating tumor DNA; PCR: Polymerase chain reaction; ddPCR: Digital droplet-PCR; qRT-PCR: quantitative real time-PCR; NSG: NOD/SCID/IL-2Rc-null mice; PDX: patient-derived xenograft.

**Table 2 ijms-23-05343-t002:** Summary of studies investigating CTCs in basic research.

Animal Model	Cancer Type	Technique	Threshold (Volume of Whole Blood)	Main Findings	Reference
Xenografts and PDX	Breast cancer (BC)	Modified CellTracks system	0.5–1.0 mL	Establishment of a method to quantify serial changes in CTC in human breast cancer xenografts and PDX.	[42]
Xenografts	Liver cancer (HCC)	Flow cytometry using DIVA software	0.5–1.0 mL	CTC detection is a predictive factor of lung metastasis. 4/11 mice developed metastasis. CTCs were only detected in these 4 metastatic mice whereas no CTCs were detected in 7/11 mice without metastasis. The amount of CTCs correlated with tumor volume, but not with the number of nodules or the largest nodule diameter.	[20]
	BC	MACS technology	100 µL	Three breast cancer xenograft models were generated, with different expression of *Stat3*. Both ctDNA and CTCs were detected, and their detection rates correlated with *Stat3* expression.	[35]
PDX	Pancreatic cancer (PDAC)	Microfluidic Chip	180–1000 µL	PDX deriving from PDAC patients were generated and divided in two groups: a group treated with an oral phosphatidylinositol-3-kinase inhibitor (BKM120) and a group treated with vehicle. Median CTCs count decreased in pre- and post-treatment in the tested group but remained unchanged in the control group.	[43]
	Colorectal cancer (CRC)	Microfluidic IMD Device	Not available	Blood samples were collected from PDX and submitted to CTCs enumeration in two groups of mice: a group treated with paclitaxel and a vehicle group. CTCs count reflected tumor burden in both groups, but the vehicle group showed higher CTCs count compared to the treated group. In addition, CTCs gradually tended towards a mesenchymal phenotype overtime.	[44]
	BC	AccuCyte®-CyteFinder® System	400–600 µL	CTCs were detected in PDX deriving from patients with breast cancer.	[45]
		Quantitative immunohistochemisty (IHC)	500–700 µL	BC-PDX models can provide a continuous and renewable source of human CTCs. There is a significant association between the presence of CTC clusters and lung metastasis potential. There is variability in CTCs number in different mice within the same PDX line, might attributed to the intratumoral heterogeneity	[46]
CDX	Breast Cancer (BC)	CellSearch System	10 mL	Analyzes identified a subpopulation of CTCs associated with the development of metastases in a xenograft assay. This subgroup of CTCs was labeled as metastasis-initiating cells (MICs), expressing EPCAM, CD44, CD47 and MET.	[47]
		CTC-iChip	20 mL	Proof-of-concept study on the feasibility to generate cell lines deriving from CTCs. A total of 6 CTCs-derived cell lines were generated. Among 5 cell lines injected in mice, three showed tumorigenic properties (BRx-07, BRx-68 and BRx-61).CTCs-derived cell lines allowed testing drug sensitivity.	[48]
		Multiparametric flow cytometry and CellSearch system	30–35 mL	CTCs from patients with TNBC were isolated and injected in xenografts, generating CDX. This permitted to interrogate transcriptomics, identifying a 597-genes signature specific of liver metastasis.	[49]
		CellSearch System	15 mL	A CDX model was developed from CTCs isolated from TNBC patient, demonstrating tumorigenicity. Characterization of CDX revealed WNT signaling as an important driver of these tumors.	[50]
	Lung cancer (SCLC)	CellSearch System	20 mL	CTCs isolated from SCLC were tumorigenic in immune-compromised mice, allowing to generate CDX. CDX outcomes and response to chemotherapy correlated with patients.	[51]
		CTC-iChip	15–20 mL	CDX were generated with CTCs isolated from SCLC patients, displaying a successful engraftment rate of 38%. CDX deriving from 1 individual but generated from CTCs isolated at different time points reliably recapitulated the drug sensitivity course of that patient.	[52]
		CellSearch System	10 mL	CDX were longitudinally generated before and upon tumor progression to test new therapeutic options in SCLC. The standard cisplatin/etoposide was compared to a new regimen with a PARP inhibitor olaparib alone or in combination with the WEE1 kinase inhibitor AZD1775 in 10 phenotypically distinct CDX. Response to therapy varied but tended to decrease when tested in CDX generated in tumor progression.	[53]
	Lung cancer (NSCLC)	CellSearch System	30 mL	CTCs were isolated from a NSCLC and injected in immune-compromised mice, allowing to generate CDX. EpCAM-dependent platform did not detect CTCs while size-based CTCs enrichment permitted to detect an abundant population of CTCs of which a majority expressed mesenchymal surface markers.	[54]
	Prostate cancer (PC) and breast cancer (BC)	CellSearch System	7.5 mL	CDX were generated with CTCs isolated from BC and PC patients. CTCs were detected in 8/8 blood samples, 6/8 bone marrow samples. In addition, human cytokeratin was detected in 6/8 harvested spleens, suggesting a persistant migratory capacity of CTCs in CDX.	[55]
	Melanoma	CellSearch System	7.5 mL	Demonstration of CTC tumorigenicity of advanced melanoma and a strategy to develop animal models when tumor material is inaccessible for PDX generation. CDX tumor growth was detected 1 month after implantation and were representative of patient tumor and treatment response.	[56]

Abbreviations: BC: Breast cancer; TNBC: triple negative breast cancer; CTCs: circulating tumor cells; CDX: CTCs-derived xenograft; PDX: Patient-derived xenograft; CRC: Colorectal cancer; ctDNA: Circulating tumor DNA; IHC: Immunohistochemistry; iMD: integrated microfluidic devices; HCC: Hepatocellular carcinoma; MACS: Magnetic activated cell sorting; PDAC: Pancreatic ductal adenocarcinoma; SCLC: Small cell lung cancer; NSCLC: Non-small cell lung cancer.

## Data Availability

Not applicable.

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
