# Peer review of "Basic Science with Preclinical Models to Investigate and Develop Liquid Biopsy: What Are the Available Data and Is It a Fruitful Approach?"

_ijms, 2022, doi:10.3390/ijms23105343_

Round 1
Reviewer 1 Report
Here, the authors give a comprehensive review on liquid biopsy studies with pre-clinical models, However, as authors are also aware and discuss the drawbacks of pre-clinical models compared to clinical patient samples (small volume accessible, not reflect translationally, animal models lack immune systems, etc), the findings through pre-clinical studies are not novel and largely have been approved by clinical studies with even larger patient cohort. Generally speaking, this review paper is lacking interests for both basic science researchers and clinicians. I would suggest the work does not reach the publication standard of IJMS and should be rejected.
minor point: language-wise, the writing is obscure and some paragraphs are a bit difficult to understand; structure wise, author should not use Result heading and restructure as main headings: general introduction, ctDNA, CTC, ctRNA/exosome and discussion, and then subheadings for ctDNA, CTC sections if resubmitting to another journal.
Author Response
REVIEWER 1
Here, the authors give a comprehensive review on liquid biopsy studies with pre-clinical models, However, as authors are also aware and discuss the drawbacks of pre-clinical models compared to clinical patient samples (small volume accessible, not reflect translationally, animal models lack immune systems, etc), the findings through pre-clinical studies are not novel and largely have been approved by clinical studies with even larger patient cohort. Generally speaking, this review paper is lacking interests for both basic science researchers and clinicians. I would suggest the work does not reach the publication standard of IJMS and should be rejected.
We would like to thank the reviewer for acknowledging the comprehensive nature of our review. Although we respect his/her opinion estimating that pre-clinical models are not novel and poorly contribute to knowledge in the field of liquid biopsy, this opinion was not shared by the large panel of experts at EACR-ESMO liquid biopsy meeting1.
minor point: language-wise, the writing is obscure and some paragraphs are a bit difficult to understand; structure wise, author should not use Result heading and restructure as main headings: general introduction, ctDNA, CTC, ctRNA/exosome and discussion, and then subheadings for ctDNA, CTC sections if resubmitting to another journal.
We followed the reviewer’s suggestion and modify the structure of the review for the headings and subheadings.
Reviewer 2 Report
In their review titled “Basic science with preclinical models to investigate and develop liquid biopsy: what are the available data and is it a fruitful approach?”, Cena et al., summarized currently available results about basic research of liquid biopsy with preclinical models. They majorly focused on cfDNA/ctDNA with 17 published works, followed by CTC studies from 17 published works and the one remaining miRNA work.
This is a very interesting review. The novelty resides in their aim, to remind society of the need for basic research with preclinical models to improve the development of liquid biopsy. There’s not much new information from each section.
Text font in Figure 1 is too small.
Author Response
In their review titled “Basic science with preclinical models to investigate and develop liquid biopsy: what are the available data and is it a fruitful approach?”, Cena et al., summarized currently available results about basic research of liquid biopsy with preclinical models. They majorly focused on cfDNA/ctDNA with 17 published works, followed by CTC studies from 17 published works and the one remaining miRNA work.
This is a very interesting review. The novelty resides in their aim, to remind society of the need for basic research with preclinical models to improve the development of liquid biopsy. There’s not much new information from each section.
We thank the reviewer for his/her very positive comments as well as for the high ratings.
Text font in Figure 1 is too small.
Absolutely. The font size of Figure 1 has been increased to enhance readability.
Reviewer 3 Report
The manuscript entitled:" Basic Science with Preclinical Models to Investigate and Develop Liquid Biopsy: What Are the Available Data and Is It a 3
Fruitful Approach? focused on a systemic revision of literature data about the adoption of liquid biopsy pre clinical models is well written and requires minor revisions to be suitable for publication:
- In the text, please, could the authors report in table 1,2 limit of detection of adopted technical procedures? In my opinion, this point may improve the readibility of the manuscript
- In the conclusion section, please, could the authors stress the concept of integrating approacch between pre clinical and clinical liquid biopsy models.
- In the recent years, an improving attention was paid to PD-L1 detection on liquid biopy determinants. Please, could the authors verify if this approach was investigated in pre clinical models?
- Liquid biopsy consists in an heterogenous concept of alterantive source of biological fluids for nucleic acids purification. According to this concept, please, could the authors also discuss about the role of alternative biological source (urine CSF) where cfDNA and CTC may be investigated?
Author Response
REVIEWER 3
The manuscript entitled:" Basic Science with Preclinical Models to Investigate and Develop Liquid Biopsy: What Are the Available Data and Is It a Fruitful Approach? focused on a systemic revision of literature data about the adoption of liquid biopsy pre clinical models is well written and requires minor revisions to be suitable for publication:
We thank the reviewer for his/her encouraging decision and for the high ratings.
In the text, please, could the authors report in table 1,2 limit of detection of adopted technical procedures? In my opinion, this point may improve the readibility of the manuscript
The small volumes of blood are indeed a specificity of this topic. We followed the reviewer suggestion and added the corresponding data in tables 1 and 2, except for CDX models (Table 2) as these studies used standard volumes of whole blood to capture CTCs in human patients, first.
In the conclusion section, please, could the authors stress the concept of integrating approacch between pre clinical and clinical liquid biopsy models.
The reviewer made a fair point; this needed to be developed. Basic and translational approaches are not in opposition. At the contrary, they may be complementary and synergize progress in the field.
This point was elaborated in the discussion (page 13):
Of importance, basic and translational approaches are not in opposition. At the contrary, they may be complementary and synergize progress in the field. Hence, studies integrating both pre-clinical and clinical materials should be valued.
In the recent years, an improving attention was paid to PD-L1 detection on liquid biopy determinants. Please, could the authors verify if this approach was investigated in pre clinical models?
Important point, indeed. Recently, investigations identified PD-1/PD-L1 pathway as an important determinant of tumor progression and allowed the development of new drugs targeting this pathway, showing great results and improving outcomes. It naturally became a target of interest in liquid biopsy as well. As an example, Winograd et al. analyzed CTCs harboring PD-L1 in patients with liver cancer and highlighted their prognostic value2.
As suggested by the reviewer, we verified whether such data in pre-clinical models were available and found one study on exosomal PD-L1 in mice models of melanoma3. Of note, it addressed PD-L1, not mRNA or cfDNA fragments coding PD-L1 but we thought it may be nonetheless relevant to mention this study. It was added to the section on ctRNA and exosomes.
The manuscript now reads (page 12):
Another study elegantly unveiled an important mechanism used by melanoma to escape immune system, using xenograft models [58]. Tumor cells upregulated PD-L1 expressed on the surface of released exosomes.
Liquid biopsy consists in an heterogenous concept of alterantive source of biological fluids for nucleic acids purification. According to this concept, please, could the authors also discuss about the role of alternative biological source (urine CSF) where cfDNA and CTC may be investigated?
Again, very pertinent comment. Liquid biopsies essentially derive from circulating analytes detected in blood but the concept is also applicable to other biological fluids such as urine, saliva, cerebrospinal fluid or bile.
This point was addressed in the discussion (page 13):
Of note, this review included data deriving from circulating analytes detected in the blood but the concept of liquid biopsy is also applicable to other biological fluids like urine, sa-liva, cerebrospinal fluid or bile. This is certainly an under-evaluated domain of liquid bi-opsy where research must also be further developed.
Round 2
Reviewer 1 Report
The authors have not finished the table 2 for blood volume of each study, I would also suggest to put animal number that undergo CTC analysis in the table 2 and how many CTCs each study find.
Author Response
This point was related to a comment made by reviewer#3 aiming to underscore the fact that preclinical models necessitate to work with very small volumes of blood. We adressed this pertinent comment and added the volumes in table 1 and table 2. In our reply, we mentioned why volumes for CDX in table 2 were not added: because these are volumes of human blood samples not animal ones. Nonetheless, we adressed this comment and added these values, as requested by the reviewer.
We would prefer not to add the number of animals used for CTCs detection and the number of detected CTCs as this would necessitate adding two additional columns in table 2, which would impair readbility.
Round 3
Reviewer 1 Report
The manuscript can be accepted.